# Ecological correlates of chimpanzee (*Pan troglodytes schweinfurthii*) density in Mahale Mountains National Park, Tanzania

**Adrienne B. Chitayat**[1]*, **Serge A. Wich**[1,2], **Matthew Lewis**[3], **Fiona A. Stewart**[2,4], **Alex K. Piel**[4]*

**1** Institute of Biodiversity and Ecosystem Dynamics, University of Amsterdam, Amsterdam, The Netherlands, **2** School of Biological and Environmental Sciences, Liverpool John Moores University, Liverpool, United Kingdom, **3** Loango Gorilla Project (Gabon), Max-Planck-Institute for Evolutionary Anthropology, Leipzig, Germany, **4** Department of Anthropology, University College London, London, United Kingdom

* chitayata@gmail.com (ABC); a.piel@ucl.ac.uk (AKP)

**Data Availability Statement:** All relevant data are uploaded to Figshare and accessible via the following URL: https://doi.org/10.6084/m9.figshare.c.5136293.

## Abstract

Understanding the ecological factors that drive animal density patterns in time and space is key to devising effective conservation strategies. In Tanzania, most chimpanzees (~75%) live outside national parks where human activities threaten their habitat's integrity and connectivity. Mahale Mountains National Park (MMNP), therefore, is a critical area for chimpanzees (*Pan troglodytes schweinfurthii*) in the region due to its location and protective status. Yet, despite its importance and long history of chimpanzee research (>50 years), a park-wide census of the species has never been conducted. The park is categorized as a savanna-woodland mosaic, interspersed with riparian forest, wooded grassland, and bamboo thicket. This heterogeneous landscape offers an excellent opportunity to assess the ecological characteristics associated with chimpanzee density, a topic still disputed, which could improve conservation plans that protect crucial chimpanzee habitat outside the park. We examined the influence of fine-scale vegetative characteristics and topographical features on chimpanzee nest density, modeling nest counts using hierarchical distance sampling. We counted 335 nests in forest and woodland habitats across 102 transects in 13 survey sites. Nests were disproportionately found more in or near evergreen forests, on steep slopes, and in feeding tree species. We calculated chimpanzee density in MMNP to be 0.23 ind/km², although density varied substantially among sites (0.09–3.43 ind/km²). Density was associated with factors related to the availability of food and nesting trees, with topographic heterogeneity and the total basal area of feeding tree species identified as significant positive predictors. Species-rich habitats and floristic diversity likely play a principal role in shaping chimpanzee density within a predominantly open landscape with low food abundance. Our results provide valuable baseline data for future monitoring efforts in MMNP and enhance our understanding of this endangered species' density and distribution across Tanzania.

**Funding:** AKP and FAS received funding from the Arcus Foundation and the UCSD/Salk Institute Centre for Academic Research and Training in Anthropogeny (CARTA).

**Competing interests:** No authors have competing interests.

## Introduction

Wildlife populations are experiencing a global decline in what has become a sixth mass extinction, a phenomenon primarily driven by human-mediated activities such as habitat destruction, overexploitation, and a rapidly changing global climate [1, 2]. Obtaining baseline data and monitoring populations over space and time are essential for guiding and evaluating the effectiveness of conservation strategies [3]. Population density and abundance estimates are useful indicators of population status [4] and capacity for long-term survival [5]. Identifying ecological factors associated with species' density can inform conservation and management bodies by helping guide the prioritization of conservation areas and enhancing our understanding of the potential consequences of environmental change.

Chimpanzees (*Pan troglodytes*) are threatened across their distribution [6], with habitat destruction and degradation, hunting, and disease as some of the leading threats to their survival [7, 8]. In Tanzania, 90% of the country's chimpanzees occur in the Greater Mahale Ecosystem (GME) where suitable habitat is being lost and fragmented by expanding human settlements, agriculture, logging, and cattle herding [8–10]. Research shows that chimpanzee density ranges from 0.1–3.7 ind/km$^2$ across sites in the GME [11, 12] and that the potential decrease in chimpanzee density between 2007 and 2014 is correlated with habitat loss [10], demonstrating the value of baseline data and repeated surveys to track population trends. Chimpanzees in savanna-woodland mosaics like the GME already live at relatively low densities (Table 1), accentuating the need to identify and protect areas critical towards chimpanzee conservation in the region.

Mahale Mountains National Park (MMNP) is the largest national park where chimpanzees in Tanzania reside and is a refugee that offers protection from common threats to them (e.g., poaching) and their habitat (e.g., settlement expansion) within the GME. While one community in the park (M group) has been the focus of long term study for decades [27], a comprehensive survey of MMNP has never been conducted, resulting in a lack of baseline data on chimpanzees distribution and density in the park. These data are crucial given the present threat of isolation and increased human disturbance the park faces from road development

**Table 1. Comparison of chimpanzee density estimates reported from surveys using nest count methodologies.**

| Location | Vegetation type | Elevation (m) | Average rainfall (mm) | Density estimate (ind/km$^2$) | Source |
|---|---|---|---|---|---|
| **Forest dominated landscapes** | | | | | |
| Budongo (Uganda) | Semi-deciduous forest | 1000–1600 | 1,620 | 1.8–1.9 | [13] |
| Gombe (Tanzania) | Tropical forest mosaic | 766–1623 | 1,495 | 2.5 | [14] |
| Kahuzi Biega (Dem. Rep. Congo) | Montane rainforest | 2030–2350 | 1,586 | 0.1 | [15] |
| Kalinzu (Uganda) | Moist evergreen forest | 1000–1500 | 1,150–1,400 | 2.8–4.7 | [16] |
| Kibale (Uganda) | Semi-deciduous forest | 1100–1600 | 1,395 | 2.4 | [17] |
| Kibira (Burundi) | Montane rainforest | 1600–2600 | > 2,000 | 0.5 | [18] |
| Nouabale-Ndoki (Republic of Congo) | Semi-evergreen forest | 330–600 | 1,728 | 1.8 | [19] |
| Nyungwe (Rwanda) | Montane rainforest | 1600–2900 | 1,744 | 0.4 | [20] |
| Odzala (Republic of Congo) | Semi-evergreen forest | 300–600 | 1,957 | 0.3–0.4 | [19] |
| Tai (Ivory Coast) | Lowland rainforest | 100–400 | 1,800 | 0.8–1.8 | [21] |
| **Open vegetation dominated landscapes** | | | | | |
| Fongoli (Senegal) | Savanna woodland mosaic | - | < 1,000 | 0.4 | [22] |
| Haut-Niger (Republic of Guinea) | Savanna woodland mosaic | - | 1,300 | 0.9 | [23] |
| Issa Valley (Tanzania) | Savanna woodland mosaic | 900–1800 | 1,200 | 0.3 | [24] |
| Mbam-Djerem (Cameroon) | Forest—woodland—savanna mosaic | 650–930 | 1,900 | 0.3 | [25] |
| Mt. Assirik (Senegal) | Savanna woodland mosaic | 100–300 | 954 | 0.1 | [26] |

and growing human settlements along its periphery, which could impact animal movement and increase human encroachment [28]. Furthermore, an investigation into the drivers of chimpanzee density and abundance in the region is lacking. Previous short and geographically restricted surveys in the park have revealed variation in chimpanzee density between some areas. However, they did not consider the effect of ecological factors [29], such as dominant vegetation type or species diversity–known to be important drivers in other populations [5, 30]. MMNP is an ideal landscape to address this topic as variation in density may arise from its immense topographic and vegetative heterogeneity. Moreover, while numerous studies have contributed on the subject of chimpanzee distribution and density patterns [7, 19, 23, 25], few have quantitatively assessed density correlates for those living in savanna-mosaics [30, 31], a habitat type often deemed marginal for the species with distinct ecological challenges (e.g., thermoregulatory stress, hydration, low fruit abundance) [31, 32].

Animal species naturally exhibit variability in their densities in response to differences in ecological variability (e.g. vegetation, topography, predation) [33, 34]. Food availability, generally influenced by vegetation structure and composition, is one of the most fundamental influences on species density, distribution, and ranging (rodents [35]; primates [36]; birds [33]; reptiles [37]), and chimpanzees are no exception [38]. As a highly frugivorous species, chimpanzees depend on the presence and distribution of fruiting trees for feeding [39, 40], as well as suitable trees for constructing nightly nests [41–43]; thus, resource abundance, especially that of fruit-bearing trees, can be used to predict chimpanzee density [44]. In particular, the abundance of fruit trees from species that provide food during periods of fruit scarcity can be one of the most critical factors influencing and limiting chimpanzee density [38] as it helps reduce the intensity of seasonal shifts in fruit availability [45, 46]. Similarly, floristic diversity can have a strong effect on chimpanzee density [5, 38, 47] when it helps chimpanzees sustain their dietary requirements throughout the year [38, 48]. For chimpanzees living in marginal habitats that often have lower overall fruit abundance and diversity and likely face more frequent or pronounced periods of resource scarcity [5], chimpanzee density may be more closely related to diversity than to overall food abundance [32, 49]. Yet, the influence of floristic diversity on chimpanzee density varies across sites, even between different savanna-mosaics [30, 49], and highlights the need for more data on this topic. Aside from the abundance and diversity of fruit trees, increased food patch size (e.g., tree size) may also help alleviate constraints from food scarcity in resource-poor areas [31], although this topic remains unexplored. The incorporation of fine-scale vegetation data into density models can assess the potential mechanisms driving variation in chimpanzee density [5, 31, 38, 50].

Chimpanzees do not uniformly utilize the landscape in time or space [51, 52]; thus, the inclusion of ecological factors related to land cover and topography, often obtained from remote-sensing data, is valuable for modeling species density and distribution [50, 53]. In open, dry landscapes, chimpanzees disproportionately rely on riparian forests for food [40, 54], nesting [26, 55], and shade [52]. Previous research in the GME suggests an association between forest cover and chimpanzee density [56]. Elevation and slope can also be important predictors of chimpanzee distribution and habitat suitability [7, 50, 57, 58] because they can influence chimpanzee nest site selection [55, 59]. However, other potentially useful and readily available topographical variables [60] remain understudied. For example, topographic heterogeneity could be valuable for predicting chimpanzee density and distribution because of the positive relationship between topographic heterogeneity and species richness [61, 62], as well as other factors like slope [60]. While chimpanzees likely respond to the availability of essential resources (e.g., food, water, nesting materials) in space and time rather than biophysical variables like percent forest cover or topographic heterogeneity, these variables can serve as insightful proxies. By incorporating both fine and broad-scale biotic and abiotic metrics within

density models, we can better understand the ecological factors associated with chimpanzee density, as well as the value of remotely sensed data necessary for large-scale predictive models.

This study examines the relationship between chimpanzee density and specific vegetative characteristics and topographical features across the MMNP landscape. To evaluate possible associations, we employed a hierarchical distance sampling (HDS) approach [63] that allows for explicit consideration of covariate influence on both the density and detection processes to more precisely model chimpanzee density patterns [64]. We predicted chimpanzee density to be higher in areas with 1) greater fruit abundance and diversity, 2) high topographic heterogeneity, and 3) more evergreen forested vegetation (includes all available forested vegetation types, i.e., riparian, lowland, and montane forests). We aim to provide baseline data on chimpanzees and their habitat (e.g., an evaluation of resource availability) in MMNP and fixed sites widely distributed across the park that can help future efforts to monitor, identify, and evaluate potential changes. MMNP is arguably the most critical area for chimpanzee conservation in Tanzania because of its size, location, and protective status; therefore, it is imperative that an assessment of this endangered species in the park (spatially) extends well-beyond the long-term research of a single community. Additionally, as a protected area, data from MMNP can serve as a point of comparison and provide insight for what to expect in the absence of human activity in the GME. For extra-park chimpanzees that face a more perilous future than those living inside park boundaries, we hope these data will allow for greater understanding of population shifts that may arise from future environmental change and better inform conservation bodies in their determination of valuable chimpanzee habitat outside of national parks.

## Methods

### Study area

This work was approved by the Tanzanian Wildlife Research Institute (TAWIRI) and the Tanzanian Commission for Science and Technology (COSTECH).

MMNP covers 1,517 km$^2$ of rugged terrain along Lake Tanganyika in western Tanzania (Fig 1). Part of the Albertine Rift, MMNP is home to numerous endemic and threatened plant and animal species [65]. The park also hosts the Mahale Mountains Chimpanzee Research Project, which along with the Gombe Stream Research Center based in Gombe NP 180km north, is one of the longest-running chimpanzee research projects in Africa, now in its 7th decade [14, 27].

MMNP is a mosaic of closed (i.e., forest) and open (e.g., woodland, grassland) vegetation types [66]. Although the northwestern region contains large blocks of continuous evergreen forest, the park is otherwise dominated by miombo and bamboo woodlands and intersected by strips of riparian forest. Elevation in the park ranges from 780–2,460 m above sea level, and the park exhibits two distinct seasons: a rainy season from October to mid-May, and a dry season, from mid-May to September.

### Study design

We collected data in MMNP from March 2018–January 2019 along 102 transects at 13 survey sites (sequentially labeled sites A–M). Considering feasibility and the average community home range sizes previously reported in MMNP [12], we determined a survey site size of 25 km$^2$. To facilitate the random site selection, we superimposed a 5 x 5 km grid over our study area, MMNP, and randomly selected grid cells (sites) using QGIS software [67]. Within each site, line transects, each 1 km long, were positioned according to a random start point and

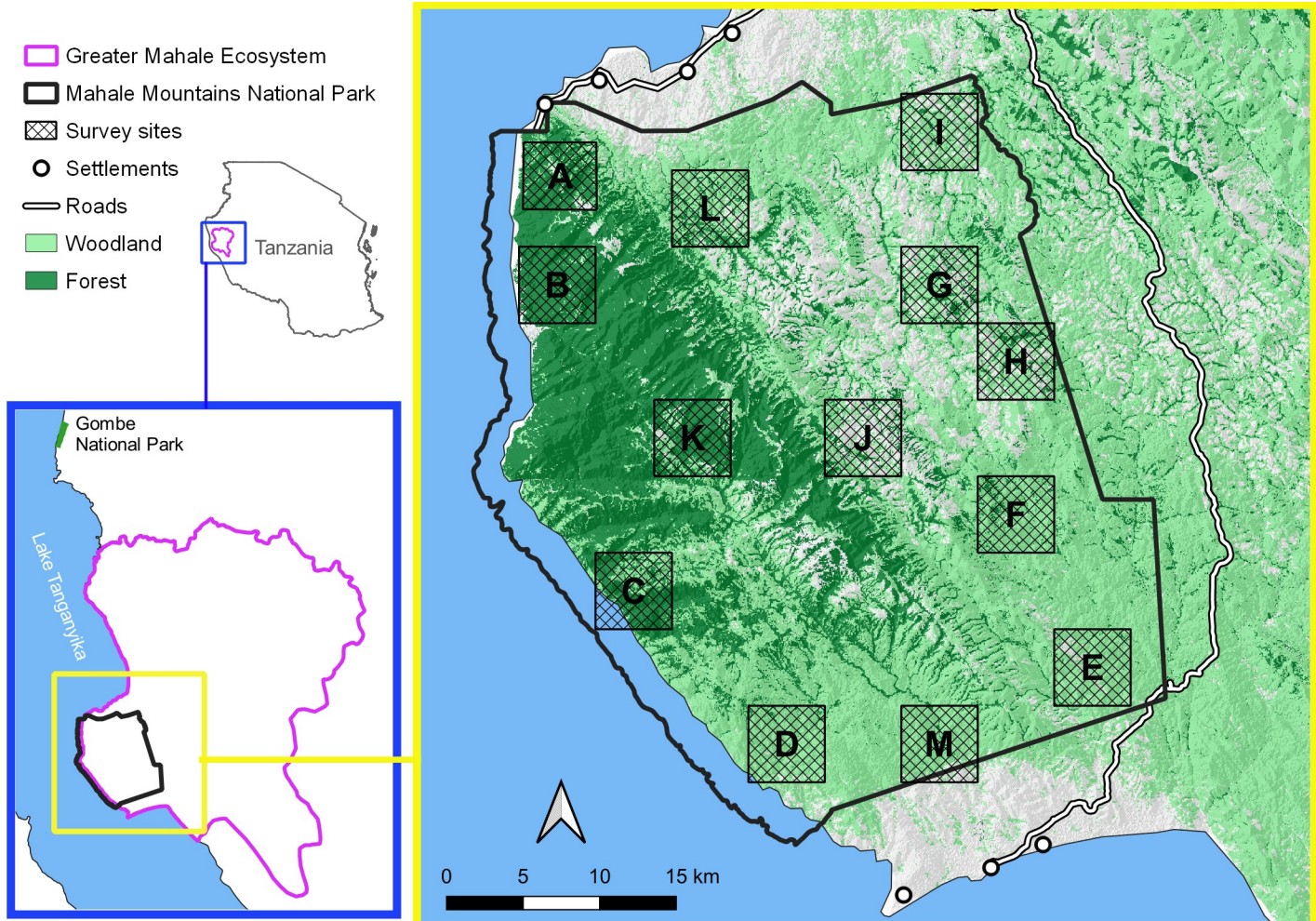

**Fig 1. Map of MMNP and its position within the GME and Tanzania.** The 13 survey sites visited during the current study are indicated with their letter name. Land cover classification courtesy of Holly E. Copeland (University of Wyoming) who sourced the data from USGS/NASA Landsat imagery.

spaced ≥1 km apart. Transects were orientated in a north to south direction, perpendicular to the drainage system.

We obtained all necessary permits from the Tanzania Wildlife Research Institute, Tanzania Commission for Science and Technology, and Tanzania National Parks and complied with all relevant regulations while conducting research within a national park and on a protected species.

### Distance sampling

Chimpanzees build nests daily for rest and sleep, allowing researchers to indirectly estimate chimpanzee density using a standing nest crop count method [13]. Walking at a pace of 1 km/ hour, survey teams recorded all chimpanzee nests observed along transects. To help us evaluate habitat conditions and the level of human encroachment in the park, we also recorded observations of human presence and activity (e.g., cut trees, snares). Following the standardized distance sampling protocol [68], we recorded the perpendicular distance between the center of each observation (e.g., nest) and the transect line using a Nikon Laser Rangefinder 550AS or

measuring tape (for distances <25m). For each observation, we recorded the GPS position, vegetation type, canopy cover (%), understory cover (%), and slope (flat, mild, moderate, steep). Vegetation types included: montane and lowland forests (forests not restricted to riparian zones), riparian forest (forests formed along watercourses), miombo woodland (discontinuous canopy of deciduous trees dominated by *Brachystegia* sp., *Julbernardia* sp., and *Isoberlinia* with grass understory), bamboo woodland (woodland with bamboo dominated understory), bamboo thicket (dense bamboo stands with scarce to no trees), wooded grassland (dominated by grasses with isolated shrubs and trees), grassland (scarce to no woody plants), and swamp. Forests were distinguished as closed or open canopy, with closed-canopy forests showing >50% canopy cover. We also recorded the tree species and age class of each nest. Nest age class was determined according to the state of nest decay based on leaf decomposition [69]: (1) leaves green and nest solid; (2) leaves wilted but nest solid; (3) some leaves lost and nest structure disintegrating; and (4) only the nest frame and <5% of leaves remaining.

## Vegetation survey

In conjunction with our chimpanzee census, we conducted a vegetation survey at each survey site with a trained botanist familiar with the plants of western Tanzania. The vegetation survey followed a belt transect design that utilized the same transects as our chimpanzee survey. We sampled five 100m x 5m plots, spaced 100m apart, along each transect. We measured and identified all trees and lianas ≥10cm diameter at breast height. We also identified and recorded vegetation type, canopy cover, and understory cover transitions continuously along transects to assess the proportion of different vegetation characteristics along each transect.

## Statistical analysis

**Predictor variables.** We determined predictor variable values at the transect level to correspond with our chimpanzee nest counts. Predictor variables derived from our vegetation survey included: forest cover (i.e., the proportion of forested habitat encountered along each transect) and several proxies of chimpanzee food availability: total basal area, mean basal area, and diversity of feeding tree species. We identified the feeding tree species that contributed to our predictors from published literature from three long-term field sites in Tanzania: Gombe [51, 70], MMNP [39], and the Issa Valley [40]. While total basal area represents overall potential food abundance, mean basal area addresses the possible influence of tree size as a food patch [31, 71]. These variables also correspond to nesting resources as chimpanzees in the GME prefer nesting in feeding species [41, 43] and large trees [59]. We calculated tree species diversity using the Shannon diversity index that accounts for the richness, relative abundance, and evenness of species [72]. We also included topographical predictors using Shuttle Radar Topography Mission satellite imagery (30 m resolution; http://earthexplorer.usgs.gov): elevation; steep slopes (proportion of slopes along each transect ≥20 degrees) [58]; topographic heterogeneity. We used terrain ruggedness to determine the degree of topographic heterogeneity, reflecting the amount of local elevation change according to the mean difference in elevation between neighboring raster cells [60]. We also included survey site as a nominal covariate to account for potential variation in nest detectability or density among sites that cannot be explained by the other variables included in our models [73].

We z-transformed all quantitative covariates to ease model convergence and achieve estimate comparability [74]. We examined the collinearity of predictor variables at the outset of our analysis using Pearson product-moment correlation coefficient and Spearman rank correlation coefficient. We considered variables highly collinear and potentially problematic when coding our models if test statistics were ≥ 0.7 or ≤ -0.7 [75]. We subsequently prevented

**Table 2. The hypothesized relationship between chimpanzee density and the covariates used to model the detection and density processes within our HDS models.**

| Habitat variables | Variable effect | Hypothesized relationship with the detection and abundance processes |
|---|---|---|
| *Detection covariates* | | |
| *Survey site* | n/a | Control for disparities that may arise from differences in the seasonal conditions experienced among sites. |
| *Forest cover* | - | Greater tree density and foliage can reduce detectability because of reduced light or obstructing/camouflaging nests. |
| *Steep slopes* | + | Steep terrain increases detectability as it leads to naturally-broken canopy [76]. |
| *Density covariates* | | |
| *Survey site* | n/a | Representative of the variability in biotic and abiotic factors between sites given that each location is a discrete area sampled. |
| *Elevation* | n/a | Possible proxy for weather conditions [77] and vegetation [66] that influence habitat use. |
| *Forest cover* | + | Forests are disproportionately used in dry landscapes, offering food [40] and nesting [55] resources. |
| *Total basal area* | + | Higher values indicate a greater abundance of food sources related to both the quantity and size of feeding tree species. |
| *Mean basal area* | + | Larger trees are generally associated with greater fruit production. |
| *Diversity* | + | Higher feeding tree species diversity can reduce the incidence of fruit seasonality and potentially offer greater resource availability in time and space [78]. |
| *Steep slopes* | + | Associated with suitable chimpanzee habitat [57, 58] and nesting sites [59]. |
| *Topographic heterogeneity* | + | Correlated with slope [60] and associated with topographical features that can influence vegetation [62] and may impact food and nesting resources. |

Covariate influence on the detection and density of chimpanzees were examined during model building and are reported as positive or negative (+/-) or not available (n/a).

highly collinear covariates from occurring in the same model [74]. We then constructed a global model of the final covariates, from which all future models were based (Table 2).

Additionally, we evaluated the overall variability of ecological factors across sites by conducting a series of Kruskal-Wallis non-parametric analysis of variance (ANOVA) tests. Moreover, we assessed the relationship between topographic heterogeneity and other ecological characteristics, such as overall species richness and slope, using p-values obtained from the Pearson correlation coefficient test in order to confirm whether the trends generally associated with topographic heterogeneity also exist in MMNP (e.g., positive correlation between topographic heterogeneity and slope). We set the alpha level to identify p-value significance at ≤0.05 for all tests.

**Hierarchical distance sampling.** We performed all analyses using R version 3.4.2 statistical software (R Core Team, 2017). We included only nests aged 1–3 in our analysis as age four nests were considered decayed [13]. We modeled observations of nests as a multinomial hierarchical coupled logistic regression [64], whereby the regression modeling the state (i.e., nest density) process is conditional on the regression modeling the detection (i.e., how animals are detected) process, accounting for imperfect detection. We applied this framework using the function 'gdistsamp' in the R package 'unmarked' [79]. Following Buckland et al. [68], we defined a truncation distance of 52 m by assessing the plotted distance frequency distribution and removing outliers from the dataset, which provide little information towards estimating the detection probability. Continuous distances were grouped into four-meter intervals to smooth heaping but retain detail. To describe nest abundance at the transect level, we used a negative binomial distribution commonly used to describe count variation in the presence of over-dispersion [64]. To verify the regression assumption of independence, we tested for spatial autocorrelation using Moran's I test [75].

We first tested and compared the performance of different detection functions (half-norm, hazard-rate) on our null model, retaining the detection function with the lowest Akaike

Information Criterion (AIC) [80]. Transect-specific covariates were then incorporated into the detection and density sub-models using a log-link function. We selected our 'best' detection model via AIC comparison and held this sub-model constant while we incorporated and compared density models. Using a combination of stepwise regression and theoretical knowledge, we tested density models and ranked them using corrected AIC (AICc) [80]. We evaluated the goodness of fit of the top-ranked model using parametric bootstrapping, simulating 1000 datasets from the fitted model, and defining a function that returned three fit-statistic (chi-square, Freeman-Tukey, sum of squares errors). For parameter estimates, we employed a multimodel based inference approach where we quantified the uncertainty that each model is the best model through the computation of model weights. We report averaged-model predictions based on models with an AICcΔ<4 as these models have greater empirical support [80]. We also calculated predictor weight on a scale of 0–1 to estimate each covariate's relative importance by summing the AICc weights for each model in which that variable appears [81] and report the significance of predictors for the top-ranked model [81].

**Conversion of nest density to chimpanzee density.** We used correction factors to convert estimates of nest density to chimpanzee density (ind/km$^2$) by incorporating nest production and decay rates (*Chimpanzee density = nest density / (nest production rate * mean nest decay rate*) [82]. We used a nest production rate of 1.1 nests/day from previous research [13], calculated according to the number of nests built per day, the proportion of nest builders, and re-use. For decay rate, we utilized all available decay rates from the GME [83–85], computed following Plumptre et al. [69]. As factors such as weather and topography affect nest decay [86], we determined the decay rate of each survey site according to location (lakeshore vs. inland), sampling season (e.g., dry vs. wet), and the proportion of open vs. closed vegetation types. As climate conditions change as one travels inland from the lake, we applied decay rates based on lake proximity. All sites within 6 km of the shoreline were considered to be within the lakeshore zone as this area encompasses lakeshore decay rate study locations [83, 84]. Lakeshore decay rates estimate 49 (dry season) and 76 days (wet season) for nests in closed vegetation and 126 days for nests in open vegetation (wet season) [83, 84]. Unfortunately, no lakeshore decay rate is available for open vegetation during the dry season, so, we calculated a rate of 167.9 days by applying the proportional difference in decay rate observed between seasons in inland open vegetation (33% increase) to the lakeshore wet season rate. For inland sites, we applied decay rates estimated by Stewart et al. [85] from the Issa Valley: 83.3 (dry) and 118.9 days (wet) for closed vegetation; 185.5 (dry) and 139.2 (wet) for open vegetation.

## Results

Transects passed through a mixture of vegetation types and consisted of 20% forested (closed-canopy 5%, open-canopy 15%) and 80% open (miombo woodland 30%, lowland bamboo woodland 30%, grassland/swamp 14%, bamboo thicket 7%) vegetation (Fig 2). Closed-canopy forests showed the greatest diversity, density, and basal area of trees ≥10cm DBH, although, for feeding tree species, miombo woodlands displayed greater species richness and diversity (Table 3). We observed minimal human presence and activity throughout the park (0.10 observations/km vs. 14.5 observations/km for wildlife), with observations recorded along only 6% of transects and at five sites. Most observations revealed only human presence (e.g., campsites, trails) and did not indicate a specific activity, although there was some direct evidence of wildlife poaching (0.01 snares/km).

We recorded 335 nests, but following truncation and the removal of age four nests, which were considered decayed according to our definition, only 263 nests were included in our analysis [69]. Of these nests, 34% were found in forests (closed-canopy 8%, open-canopy 26%) and

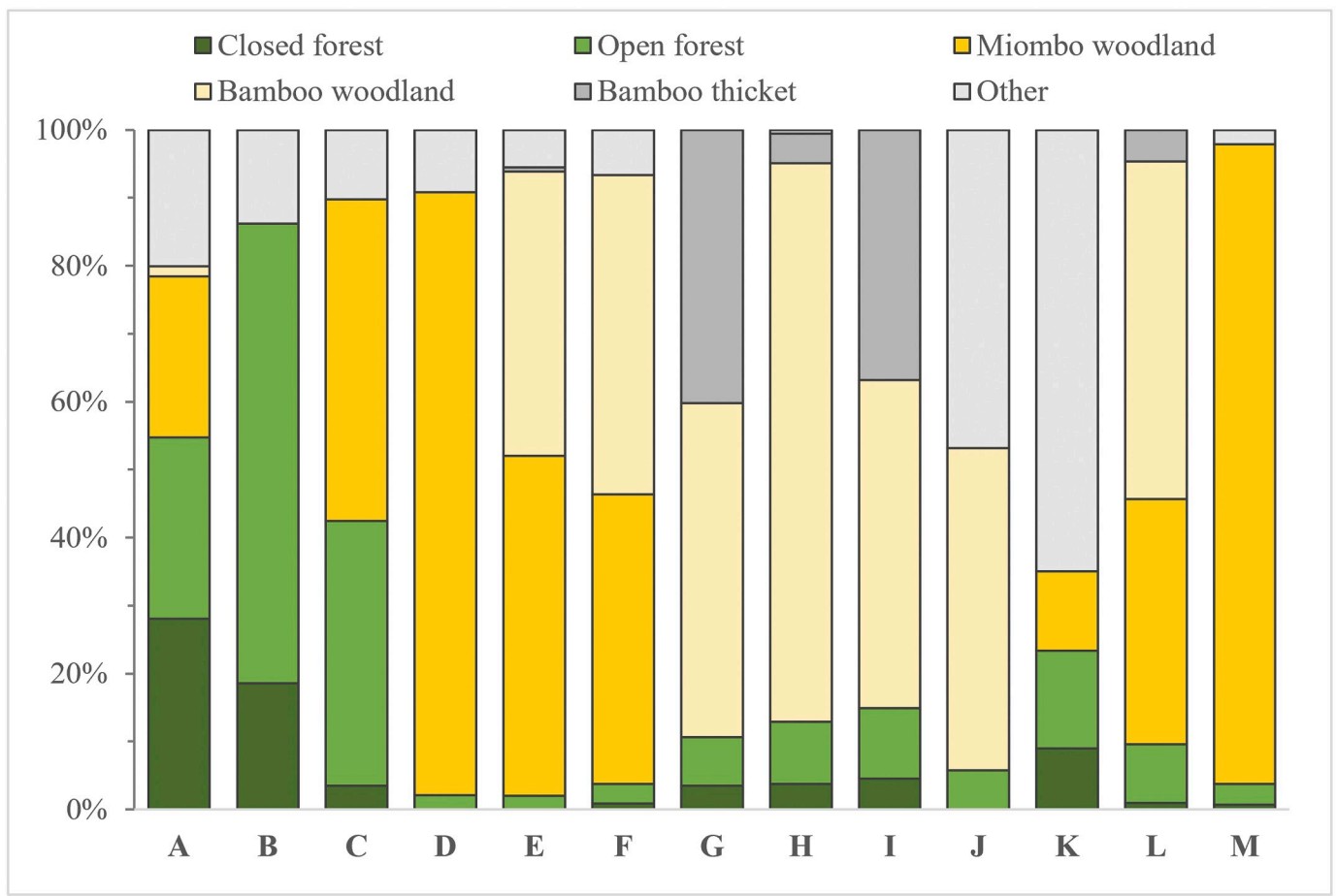

**Fig 2. Graph showing vegetation type percentages observed at each site.** We distinguish forests according to canopy cover (open forest <50% coverage; closed-canopy ≥ 50%). "Other" includes non-wooded vegetation types (e.g., grassland, swamp).

66% in open vegetation (miombo 46%, bamboo 19%, wooded grassland 1%). Nests were disproportionately observed on steep slopes (≥20 degrees), with 56% of nest locations found on steep slopes even though steep slopes accounted for only 14% of transect terrain. We found nests in >33 tree species, but we observed the majority (51%) in only four species: *Julbernardia*

**Table 3. Overview of important vegetation types found in MMNP and utilized by chimpanzees for nesting.**

|  | Open-canopy forest | Closed-canopy forest | Miombo woodland | Bamboo woodland |
|---|---|---|---|---|
| *All species* |  |  |  |  |
| No. of species | 95 | 95 | 149 | 19 |
| Diversity (Shannon Index) | 3.5 | 4.7 | 3.8 | 2.4 |
| Tree density/ha | 76 | 237 | 190 | 5 |
| Basal area/ha | 4.9 | 14.5 | 3.8 | 2.4 |
| *Feeding species* |  |  |  |  |
| No. of species | 40 | 52 | 56 | 10 |
| Diversity (Shannon Index) | 3.0 | 3.6 | 5.3 | 1.3 |
| Tree density/ha | 38 | 122 | 118 | 3 |
| Basal area/ha | 2.7 | 6.6 | 5.6 | 0.1 |

**Table 4. Chimpanzee density estimates with 95% confidence interval (CI) and mean covariate values for each sample site.**

| Site | Topographic heterogeneity | Forest cover | Total basal area ($m^2$/ha) | Mean basal area ($m^2$/ha) | Shannon diversity index | Chimpanzee density (ind/$km^2$) | Chimpanzee density (ind/$km^2$) 95% CI |
|------|---------------------------|--------------|-----------------------------|----------------------------|-------------------------|----------------------------------|------------------------------------------|
| A | 9.13 | 57% | 10.60 | 0.05 | 2.00 | 3.43 | 1.36–8.67 |
| B | 7.23 | 86% | 11.91 | 0.07 | 2.13 | 3.24 | 1.23–8.74 |
| C | 4.62 | 42% | 7.22 | 0.04 | 2.34 | 0.54 | 0.30–0.97 |
| D | 3.93 | 2% | 6.89 | 0.05 | 1.22 | 0.10 | 0.06–0.20 |
| E | 3.09 | 2% | 4.51 | 0.04 | 1.75 | 0.08 | 0.04–0.14 |
| F | 5.46 | 4% | 6.52 | 0.07 | 1.56 | 0.20 | 0.12–0.32 |
| G | 5.51 | 11% | 1.42 | 0.06 | 0.97 | 0.11 | 0.06–0.21 |
| H | 4.44 | 13% | 1.10 | 0.04 | 1.33 | 0.09 | 0.05–0.16 |
| I | 3.57 | 15% | 3.27 | 0.04 | 1.63 | 0.09 | 0.05–0.16 |
| J | 5.55 | 5% | 0.48 | 0.01 | 1.35 | 0.11 | 0.06–0.20 |
| K | 7.12 | 23% | 2.98 | 0.04 | 1.39 | 0.39 | 0.23–0.65 |
| L | 5.79 | 10% | 5.31 | 0.06 | 1.89 | 0.34 | 0.22–0.53 |
| M | 4.04 | 4% | 7.47 | 0.07 | 1.95 | 0.21 | 0.12–0.36 |

*globiflora*, *Brachystegia spiciformis*, *B. bussei*, and *Xylopia parvaiflora*. Feeding tree species accounted for 78% of the nesting species utilized by chimpanzees in MMNP and 94% of all nesting trees we recorded. We identified at least 259 different species of trees during and vegetation survey, of which 83 species are used for feeding by chimpanzees [39, 40, 50, 70] (S1 Table).

We found that steep slopes and topographic heterogeneity were highly correlated ($r_p <$ -0.93, df = 100, P < 0.001) and coded models accordingly. We also found a significant positive correlation between topographic heterogeneity and overall tree species richness ($r_p =$ 0.20, df = 100, P = 0.05), but not feeding tree species ($r_p <$ 0.001, df = 100, P = 0.98). Non-parametric ANOVA tests revealed significant differences among sites for all ecological characteristics considered in our models (*elevation*: $F_{2, 12} =$ 80.6, P < 0.001; *forest cover*: $F_{2, 12} =$ 59.8, P < 0.001; *total basal area*: $F_{2, 12} =$ 72.5, P < 0.001; *mean basal area*: $F_{2, 12} =$ 46.1, P < 0.001; *diversity*: $F_{2, 12} =$ 53.0, P < 0.001; *steep slopes*: $F_{2, 12} =$ 30.1, P = 0.003); *topographic heterogeneity*: $F_{2, 12} =$ 41.0, P < 0.001) (Table 4). Moran's I test confirmed the independence of our samples, showing no spatial autocorrelation between nest counts (Moran's I = 0.04, p = 0.02).

## Covariate influence on density and detection

The hierarchical modeling approach that we applied allowed us to derive a model that performed better than null models that did not consider covariate effects on detection or density. The results of our bootstrapped goodness of fit test confirmed that our top-ranked model exhibited good fit with our data (Chi-square: $x^2 =$ 0.26; Freeman-Tukey: q = 0.23; sum of squares: SSE = 0.27). Of the models we tested, only eight models contributed towards the cumulative AICc weight (Table 5), and predictor weights differed considerably in their relative importance and contribution towards density estimates: topographic heterogeneity (0.98), total basal area (0.63), diversity (0.55), forest cover (0.53), mean basal area (0.16), steep slopes (0.03), elevation (0), site (0). Covariates included in the top-ranked model exhibited a significant effect on chimpanzee density (topographic heterogeneity: p < 0.001; *total basal area*: p < 0.001) (Fig 3). Our results estimate chimpanzee density at 0.23 ind/$km^2$ (0.16–0.35 95% CI) across all MMNP, but estimates varied significantly among sites ($F_{2, 12} =$ 58.23, P < 0.001), ranging from 0.09–3.43 ind/$km^2$.

**Table 5. The weight and AICc value of each model contributing to our chimpanzee density predictions.**

| Model | AICc | Model weight | Cumulative weight |
|---|---|---|---|
| λ(heterogeneity[a] + TBA[b]) p(site) | 751.44 | 0.21 | 0.21 |
| λ(heterogeneity + forest cover + diversity[c]) p(site) | 751.89 | 0.17 | 0.38 |
| λ(heterogeneity + forest cover + TBA) p(site) | 752.00 | 0.16 | 0.53 |
| λ(heterogeneity + TBA + diversity) p(site) | 752.37 | 0.13 | 0.66 |
| λ(heterogeneity + MBA[d] + diversity) p(site) | 752.52 | 0.12 | 0.79 |
| λ(heterogeneity + forest cover + TBA + diversity) p(site) | 753.33 | 0.08 | 0.87 |
| λ(heterogeneity + forest cover) p(site) | 753.95 | 0.06 | 0.93 |
| λ(heterogeneity + forest cover + TBA + MBA + diversity) p(site) | 754.6 | 0.04 | 0.97 |

All models include our best detection sub-model (p) but vary by density sub-model (λ).

[a] Topographic heterogeneity

[b] Total basal area

[c] Shannon diversity index

[d] Mean basal area

## Discussion

MMNP is home to one of the longest-running research studies of any single chimpanzee community [27]. Yet, in>50 years of research and 35 years since the park's creation, there was no park-wide census of one of its most charismatic speciesuntil the current study. Given the park's protective status, limited human encroachment, and that it's located within the GME where the greatest number of Tanzania's chimpanzees occur, MMNP is a key area for chimpanzee conservation. Chimpanzees were present throughout the ecologically diverse park and we found that characteristics related to food and nesting resources are strongly associated with chimpanzee density, resulting in significantly variable densities that ranged from 0.09–3.43 ind/km$^2$ among 13 sites.

### Vegetation type

Our results are consistent with those from other chimpanzee surveys (see Table 1), showing that across the savanna-woodland mosaic of MMNP, chimpanzees exist at a relatively low density of 0.23 ind/km$^2$. Like other sites dominated by open vegetation [23, 25, 30], forests are an important vegetation type in MMNP. Our results show that chimpanzees disproportionately use forests for nesting and that there is a positive association between forest cover and chimpanzee density. Sites located in the park's northwestern region exhibit the most forest cover and the highest chimpanzee densities (e.g., site B = 3.24 ind/km$^2$), with densities 6–38 times greater than woodland dominated sites that characterize the remainder of the park. These findings support observations from previous researchers that this region of MMNP hosts a high density of chimpanzees [29], which they largely attributed to high food availability [87]. Our study provides empirical support for this assertion by demonstrating that the northwest region hosts the greatest basal area of feeding species. Furthermore, Site B coincided substantially with the home range of M-group. Based on the direct identification of community members, M-group density has varied over the years, ranging from 2.6–3.7 ind/km$^2$ from 1996–2012 [12] and 3.5 ind/km$^2$ during the study period. Similarities between these independent metrics of density validate our methodology and analysis for estimating chimpanzee density.

In a primarily open landscape, non-forested vegetation types inevitably provide crucial resources for chimpanzees (Fig 4). Regionally, chimpanzees derive much of their food [40, 54] and nesting species [41, 43] from miombo woodlands, and several results from our study

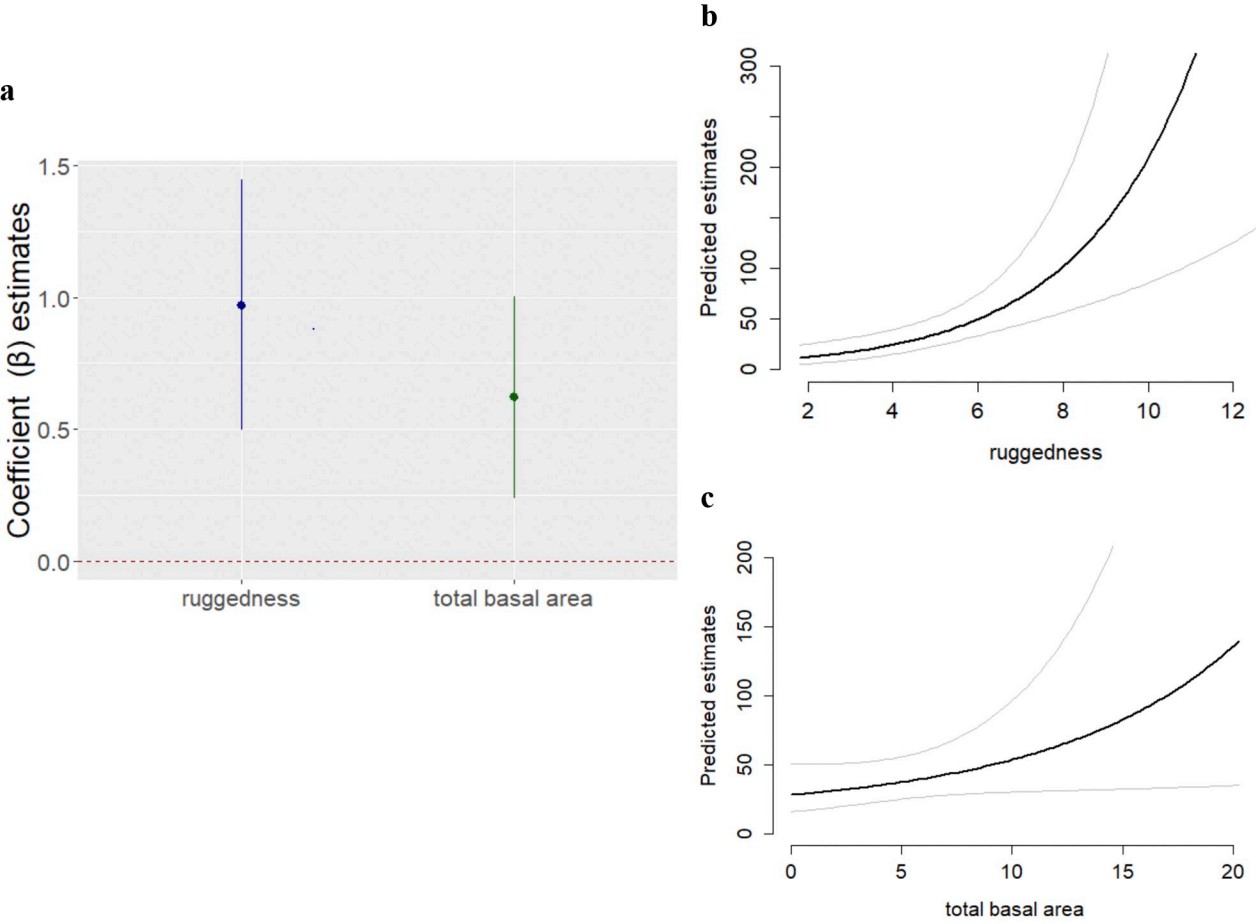

**Fig 3. Predictor variable plots from the top-ranked model of nest density.** (a) Plot of coefficient estimates (circles) presented with 95% CI (vertical lines), confirming their significance (because CI does not cross zero); (b) response curves of predicted nest density against topographic heterogeneity and (c) total basal area.

indicate the value of this vegetation type in MMNP. For example, chimpanzee density seems to fluctuate with the availability of miombo woodland when survey sites have the same amount of forest cover, e.g., site G (10% forest, 0% miombo, 0.11 ind/km$^2$) vs. site L (11% forest cover, 34% miombo, 0.34 ind/km$^2$). Additionally, chimpanzee density was positively associated with the basal area and diversity of feeding tree species, reflecting the importance of species-rich habitats like miombo woodlands that display a comparatively high diversity and abundance of feeding tree species. This contrasts findings from the savanna-forest mosaic of Lagoas de Cufada Natural Park (Guinea-Bissau), where chimpanzee nest abundance was negatively correlated with the basal area of food plant species that is indicative of dense forests. The relatively greater importance of basal area than forest cover showcases the necessity of resources across the landscape. These results are likely driven by the highly seasonal nature of the GME [27, 40, 88] that results in the variable use of different vegetation types over the year. Previous research describes chimpanzee reliance on woodlands during the dry season when forest fruits are less abundant [40]. Moreover, the density of feeding tree species in MMNP (5.3 m$^2$/ha, SD = 3.5) is low in comparison to other chimpanzee sites where similar data are available, e.g., Kibale National Park (Uganda) (7.6–9.9 m$^2$/ha for top 10 fruit species only) [5], and likely compels chimpanzees to seek resources wherever available. Therefore, areas with a diversity of

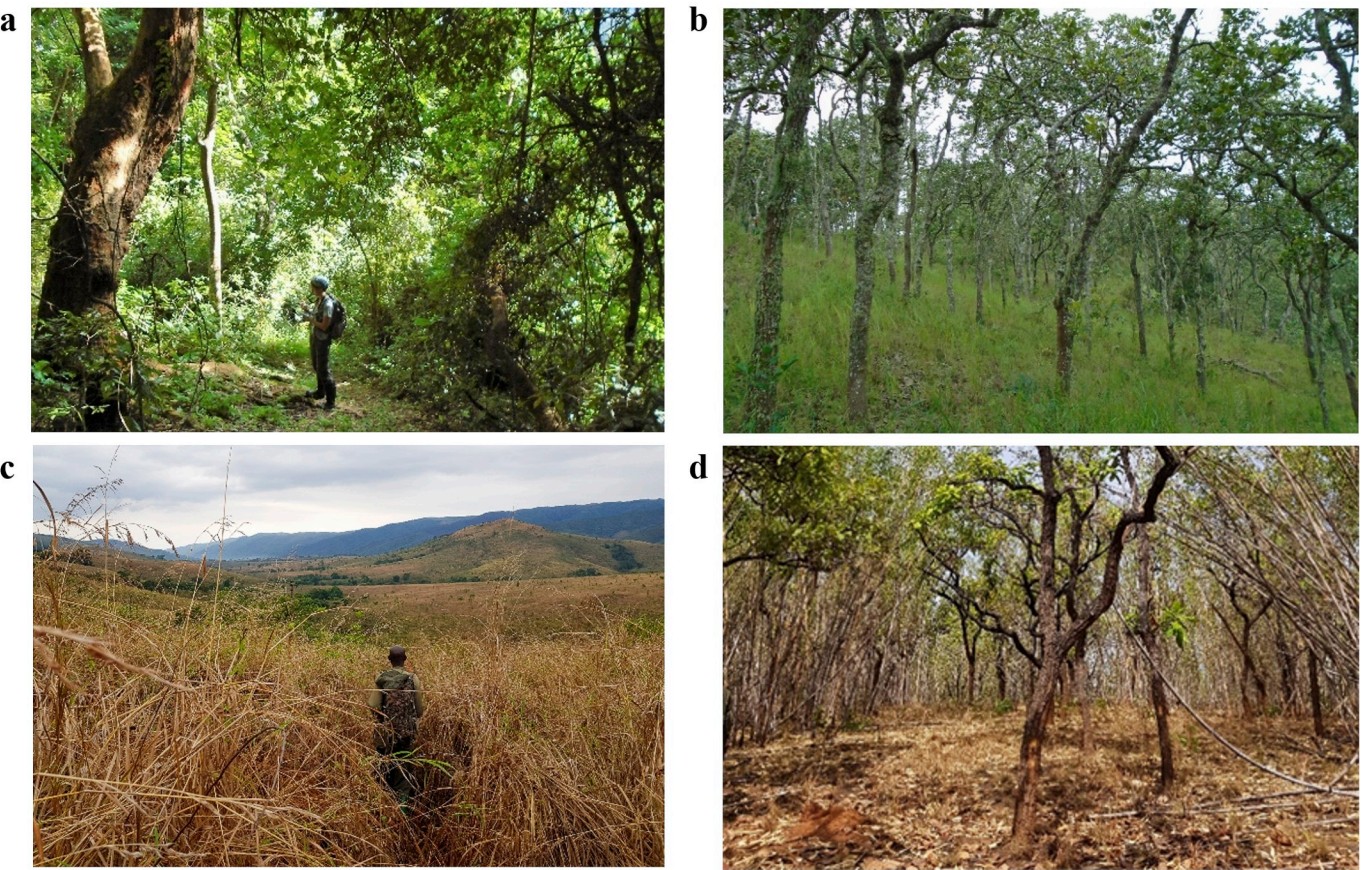

**Fig 4. Selection of vegetation types observed in MMNP, illustrating its mosaic landscape.** (a) lowland closed canopy forest; (b) miombo woodland; (c) grassland; (d) lowland bamboo woodland (Photos courtesy of A.C.).

vegetation types (Fig 4) capable of supplying a greater abundance and diversity of resources are likely advantageous for chimpanzees in MMNP.

## Fruit availability

Our results support similar findings from other locations [19, 38, 89] that floristic differences between sites play a pivotal role in density variability. In addition to the significant, positive effect of total basal area, the mean basal area of feeding trees had a positive, albeit relatively weak, effect on chimpanzee density in our models and demonstrates the value of large food trees, with presumably greater amounts of food, for chimpanzees living in a primarily open landscape. However, this finding may also be influenced by chimpanzee preference for nesting in large trees, as shown by previous research conducted in the GME [43, 59]. Our analysis also confirmed the importance of floristic diversity for this species in MMNP, with our models demonstrating a positive correlation between chimpanzee density and feeding tree species diversity. In addition, our results showed that topographic heterogeneity, the most important predictor in our models, adheres to the positive trend generally shown between heterogeneity and species richness [61]. This suggests the importance of species-rich areas for chimpanzees, which may also provide diverse resources from food items not

analyzed during this study (e.g., herbaceous growth, insects). The importance of diversity for chimpanzees in MMNP, compared to other chimpanzee sites, may be the result of both necessity and functionality. Resource diversity may be more valuable for chimpanzees living in low fruit abundance areas like MMNP, where an ability to diversify their diet allows individuals to compensate for low food density and maintain their nutritional needs. Potts et al. [47] examined two adjacent communities in Kibale and found that Kanyawara chimpanzees (who live at a lower density than their Ngogo neighbors) demonstrate greater dietary diversity than Ngogo chimpanzees that live in an area with a significantly greater abundance important food species. For frugivorous animals, floristic diversity is advantageous when it reduces the fluctuation of fruit availability across seasons [89]. Thus, a diversity of plants that all produce fruit simultaneously is not functionally equivalent to species diversity that helps diminish fruit scarcity, e.g., via asynchronous fruiting [38]. That MNNP has comparatively low food availability likely explains the importance of feeding tree species abundance, size, and diversity towards chimpanzee density. Future research that includes an investigation into the phenology of chimpanzee food resources is necessary to evaluate if and how chimpanzee density shifts with the availability of different resource functional classes (e.g., fallback food).

## Nesting trees

The ecological characteristics of sleeping sites inherently drive our assessment of chimpanzee density patterns in MMNP due to our use of chimpanzee nests for our analysis. The significant correlation between total basal area and chimpanzee density is, therefore, likely related to nesting resources and not only food. In the savanna-woodland mosaic of MMNP, where tree density is low compared to forest-dominated sites, chimpanzees may strategically utilize feeding species. Nesting in feeding trees may help individuals reduce travel costs and energy expenditure [90] and defend key resources from frugivorous competitors [42]. Likewise, as topographic heterogeneity is positively correlated with slope, the significance of this variable in our models is likely partially driven by our finding that chimpanzees in MMNP prefer to nest on steep slopes. Chimpanzee preference for nesting on steep slopes is unlikely to be the byproduct of where preferred nesting trees are located since most trees from nesting species (69%) were not found on steep slopes. Instead, a preference for nesting on steep slopes may reflect an alternative motivation, such as vocal communication [91], or predator defense as steep slopes may provide a better view of the surrounding habitat and taller trees [55, 59]. Large carnivores, such as leopards (*Panthera pardus*) and lions (*P. leo*), are found across MMNP (Chitayat, unpublished data) and the GME [92], and are a well-documented threat to chimpanzees [93, 94]. Yet, steepness was a relatively unimportant predictor in our models (predictor weight = 0.03), especially in comparison to topographic heterogeneity, whose association with density extends beyond chimpanzee preference for nesting on steep slopes. Research regarding the impact of predation pressure on chimpanzee density and distribution is needed for greater clarification. Future models could benefit from the incorporation of additional ecological predictors like predator density and other factors that may impact sleeping site selection, such as proximity to water sources [55] and microclimate [85]. Moreover, because our research was limited to one visit per survey site, we could not assess the seasonal effects often reported to influence chimpanzee nesting patterns, habitat use, and ranging within the GME [12, 40, 43, 55]. Future research would benefit from collecting data during both the wet and dry seasons to determine if the patterns we observed in this study are consistent across the annual cycle.

## Conclusions

Our study offers the first comprehensive density data on chimpanzees within a key conservation area in Tanzania. Our results show that survey site estimates are highly variable and dependent on the to ecological conditions of the site, with topographic heterogeneity, forest cover, and food availability demonstrating positive associations with chimpanzee density across the MMNP landscape. With this information, conservation and management bodies are better equipped to identify and prioritize suitable chimpanzee habitat within the GME. For instance, based on our finding that site-wide food availability is more important than forest cover availability, we recommend that conservation practitioners take a landscape approach that considers the importance of species-rich habitats and overall habitat diversity, particularly the availability of miombo woodlands. Moreover, our data do not assess the full extent of these chimpanzees' range but instead, where they sleep, which can be up to nine km from where they range during the day, as observed at Issa Valley (personal communication). Thus, we recommend that conservation practitioners consider other chimpanzee habitat use indicators, e.g., travel paths [95], habitat connectivity [58], to encapsulate the habitats necessary for their continued survival fully.

Outside the park, the destruction and degradation of habitat from human activities threaten chimpanzee viability across western Tanzania by altering habitat composition and availability and, consequently, chimpanzee resources and connectivity [53, 58, 96]. This threat is compounded by land conversion for agriculture that often occurs close to rivers where riparian forests are found. Additionally, while we are encouraged by the limited anthropogenic activity we observed in MMNP, present threats just outside the park (e.g., road development, urban expansion, and growing human population size) that place even protected areas at risk [9], threatening them with human encroachment and eventual isolation. Additionally, the SARS-CoV-2 pandemic may exacerbate conservation threats if it results in reduced funding for protected areas and an increase in poverty that places greater pressure on the park [97]. The pandemic's associated illness (COVID-19) also brings into sharper focus the risk of disease transmission our closest living relatives face when living in close proximity to humans. To track potential changes in chimpanzee density and their habitat, we recommend re-visiting MMNP survey sites, and extra-park locations, at regular intervals (at least every five years) in accordance with Tanzania's national chimpanzee conservation action plan [98]. We hope our results from MMNP can serve not only as a baseline for MMNP but a point of comparison for the region to help researchers identify the impacts of human activities more precisely outside of the national park. Chimpanzees are a resilient species and can persist successfully in human-modified landscapes [26, 51, 56] when they are not directly exploited through hunting and appropriate conservation actions are taken to promote their longevity [99]. Through continued monitoring efforts and the development of well-informed management strategies that do not only *react* to population declines but adequately anticipate population vulnerability, we can hopefully ensure the long-term persistence of chimpanzees in the GME and Tanzania.

## Supporting information

**S1 Table. Feeding species recorded in MMNP during our vegetation survey and identified from previous literature from Tanzania: M = Mahale [39]; G = Gombe [50, 70]; I = Issa Valley [40].**
(DOCX)

## Acknowledgments

We would like to thank the Greater Mahale Ecosystem Research and Conservation (GMERC) and the Frankfurt Zoological Society, who partnered with us on this project. We also thank the

UCSD/Salk Institute Center for Academic Research and Training in Anthropogony (CARTA) that supports GMERC and the permanent research station at Issa Valley. We are grateful to the Tanzania Wildlife Research Institute (TAWIRI), Tanzania Commission for Science and Technology (COSTECH), and Tanzania National Parks (TANAPA) for their permission to conduct this research and support during the fieldwork. Thank you to Godfrey Stephano, Makasa Mlekwa, Gabriel Laizer, Moses Anyelwisye, Shabani Kabangula, Mustafa Kizimba, and numerous others for their invaluable assistance and expertise in the field, without whom this research would not have been accomplished. Finally, a special thank you to Edward Kohi, Anja Hutschenreiter, Maria Voigt, Dave Seaman, and Andrew Royle for taking the time to provide advice and guidance on our statistical analysis, and Holly E. Copeland for providing the GME landcover layer that greatly improved our ability to assess spatial data.

## Author Contributions

**Conceptualization:** Adrienne B. Chitayat, Fiona A. Stewart, Alex K. Piel.

**Data curation:** Adrienne B. Chitayat.

**Funding acquisition:** Fiona A. Stewart, Alex K. Piel.

**Investigation:** Adrienne B. Chitayat, Matthew Lewis.

**Methodology:** Matthew Lewis, Fiona A. Stewart, Alex K. Piel.

**Project administration:** Adrienne B. Chitayat, Fiona A. Stewart.

**Supervision:** Serge A. Wich, Alex K. Piel.

**Writing – original draft:** Adrienne B. Chitayat.

**Writing – review & editing:** Adrienne B. Chitayat, Serge A. Wich, Matthew Lewis, Fiona A. Stewart, Alex K. Piel.

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
