## [Decision Letter · Decision Letter 0]

5 Nov 2020

PONE-D-20-30689

Ecological correlates of chimpanzee (Pan troglodytes schweinfurthii) density in Mahale Mountains National Park, Tanzania

PLOS ONE

Dear Dr. Piel,

Thank you for submitting your manuscript to PLOS ONE. After careful consideration, we feel that it has merit but does not fully meet PLOS ONE’s publication criteria as it currently stands. Therefore, we invite you to submit a revised version of the manuscript that addresses the points raised during the review process.

We look forward to receiving your revised manuscript.

Kind regards,

Bi-Song Yue, Ph.D

Academic Editor

PLOS ONE

3.We note that [Figure(s) 1] in your submission contain map images which may be copyrighted. All PLOS content is published under the Creative Commons Attribution License (CC BY 4.0), which means that the manuscript, images, and Supporting Information files will be freely available online, and any third party is permitted to access, download, copy, distribute, and use these materials in any way, even commercially, with proper attribution. For these reasons, we cannot publish previously copyrighted maps or satellite images created using proprietary data, such as Google software (Google Maps, Street View, and Earth). For more information, see our copyright guidelines: http://journals.plos.org/plosone/s/licenses-and-copyright.

1.    You may seek permission from the original copyright holder of Figure(s) [1] to publish the content specifically under the CC BY 4.0 license. 

Reviewers' comments:

Reviewer's Responses to Questions

**Comments to the Author**

1. Is the manuscript technically sound, and do the data support the conclusions?

Reviewer #1: Yes

Reviewer #2: Partly

2. Has the statistical analysis been performed appropriately and rigorously? 

Reviewer #1: I Don't Know

Reviewer #2: Yes

3. Have the authors made all data underlying the findings in their manuscript fully available?

Reviewer #1: Yes

Reviewer #2: No

4. Is the manuscript presented in an intelligible fashion and written in standard English?

Reviewer #1: Yes

Reviewer #2: Yes

5. Review Comments to the Author

Reviewer #1: This is a good data set that is well analyzed. I think that with some work it is definitely suitable for publication in PLoS ONE. However, I have two major criticisms that need to be addressed before I recommend publication.

First, the introduction is lacking in terms of setting up the motivation for the study. The authors make a big deal in the intro of the importance of estimating the overall pop density of chimps in Mahale Park as a benchmark for future conservation efforts. This is reinforced by Table 1, where the estimate densities of other important chimp populations. I was thus led to believe that this was going to be a major goal of the paper. But when it comes to the actual results, all we get is this: Line 303: “Our results

estimate chimpanzee density at 0.23 ind/km2 (0.16 – 0.35 95% CI) across all MMNP, but estimates varied significantly among sites (F2, 12= 58.23, P < 0.001), ranging from 0.09 – 3.43 ind/km2 (Table 306).” This result isn’t even mentioned in the discussion. Do the authors really believe this 0.23 ind/km2 (0.16 – 0.35 95% CI) estimate is robust enough to serve as a benchmark to evaluate the effectiveness or lack thereof of conservation measures at Mahale Park. If yes, explain why and defend that position; if not, drop this angle from the intro.

The second major goal of the study, determining the specific ecological predictors associated with chimp density at Mahale Park, also lacks a good set up. Lines 92-95: “We predicted chimpanzee density to be higher in areas with 1) greater fruit abundance and diversity, 2) high topographic heterogeneity, and 3) more evergreen forested vegetation (includes all available forested vegetation types, i.e., riparian, lowland, and montane forests).” I think the authors need to spend some more time in their introduction setting up the motivation for these predictions. Science is about filling gaps in knowledge, setting up tests that make us favor specific hypotheses/theories over others. The authors don’t do a great job explaining how the answers to these specific questions fit into some bigger debate. Basically, what is at stake here with the specific answers to these questions? What previous research and debates are being built upon, and how will the specific answers given in this paper contribute to the resolution of those debates? There is a little bit of this in the discussion, but those sorts of things should be raised more in the intro. As of now, the intro is too broad, focused much more on the very general issue of trying to convince the reader that it is important to know about ecological predictors of chimp density, and not enough on the more specific issues of what we already know, what we don’t know, and why we need to do X, Y or Z to resolve these issues.

Finally, my second major criticism is that a lot of the discussion is too far off-topic. In some places it reads more like the authors personal opinions about the best way to conserve chimps in the Mahale area. This is all fine and reasonable stuff, but it doesn’t have a direct relationship to what is analyzed in the paper.

Below are some notes I made while reading the paper that I hope the authors find useful:

Line 95: Baseline data on what? On overall population size in Mahale park? Or are you taking about three 3 predictions in the previous sentence?

Line 139-141: But you don’t mention anything about human disturbance in the intro. If you are actually going to analyze this in the paper, you need to set it up in the intro. If you are not, you don’t need to describe it in the methods (i.e., you don’t need to describe your entire methodology, but only the bits of it are relevant to the specific questions addressed in the paper).

Line 175: by ‘individual trees’, do you mean ‘different tree species’?

2.5 Statistical analyses. You said you had clusters of 5-9 transects per site. Were these transects considered as statistically independent predictors n the model? I worry about autocorrelation, as is suggested by the clustering term. Please clarify.

Line 186-187: “We built models to determine the effect of predictor variables on the partially observed true state (i.e., nest density) and detection (i.e., how animals are detected) processes.”

-I don’t understand what these two things you are predicting are exactly, and how they differ, Please explain a bit more.

Table 3: Do any of the sites correspond to the location of the habituated Mahale chimp groups? I ask because it would be nice to get a sense of the validity of your estimates by knowing how well they compare to the most accurate and precise way of getting densities (i.e., habituating chimps, individually identifying them, and counting them). Many of the density estimates shown in Table 3 have suspiciously narrow confidence intervals.

Conclusions: Sure, connectivity is great; I am all for connectivity. But what does connectivity have to do with the specific questions you addressed in this paper? You could have also wrote about how poaching, or the chimp pet trade, are bad for chimp populations, and we should reduce those things. But these things too don’t have much of a direct connection with the actual results of your paper.

Line 454-468: Again, I agree with all of these things you write here, but what do they have to do with the results of your paper? In your paper, you found X, when it could have turned out that you found Y. Your discussion should deal with the greater significance/meaning of the fact that you found X instead of finding Y. These things that you are writing about here have little to do with either X or Y.

Line 469-486: Same comment as above.

Reviewer #2: This was a well-written and interesting manuscript that applied sophisticated statistics and extensive field work to the question of estimating chimpanzee densities in different habitats across Mahale Mountains National Park. I believe that the manuscript provides a useful set of results that can inform conservation efforts in the Greater Mahale Ecosystem and within the park itself. I do have several minor issues that I would like to see the authors address.

First, the conclusion section does not refer much to the results from the study. Indeed, there is only one sentence in the conclusion section that refers directly to the results of this study (lines 440-443). Proposed follow-up research is also briefly mentioned (line 469 ff.), but I think it would be valuable explain more clearly in the conclusion how the results of this study, or alternatively the results of the proposed follow-up research, can assist in conservation efforts beyond general statements about biodiversity surveys etc. This would more clearly link the conclusions with the data.

Another issue that I think would strengthen the conclusions in the manuscript: it would be valuable for the reader to discuss the relationship between chimpanzee community range sizes and daily travel distances in Mahale, and the 25 km^2 study sites. You note at the end that you're sampling where chimpanzees sleep, not their entire range (obviously beyond the scope of this project). while 25 square kilometers is a large area, and one would not expect chimpanzees to routinely travel 5 km to nest, I have occasionally seen chimpanzees traveling a kilometer or two at the end of the day to a nesting site, so the question of overall space use vs nest site selection seems quite relevant. Was the size of each study site (5km x 5 km) chosen with the ranging patterns of chimpanzees in mind? It would at least be useful to expand on the difficulties involved in inferring overall space use from sleeping sites. Community range data would be especially useful because it is unlikely that chimpanzees will nest in contested territory between two community ranges, regardless of food species abundance and whether chimpanzees are entering the area during the day to feed (presumably in large groups). I don't know if you have access to those data, and the selection of 13 randomly-chosen sites makes it less likely that an unlucky site placement will bias your results, but it would help the reader and strengthen the conclusions to include some discussion of chimpanzee behavior.

Methods:

I wasn't totally convinced by the decision to use p values to exclude one of two co-linear terms from the full model. Would it be possible to simply compare alternative models using each of the two terms that were colinear in the full model (steep slopes and ruggedness), rather than excluding one using p values in the full model?

In addition, I had several other small suggestions:

Line 160: "accomplished within 1 - 10 of the chimpanzee data collection" - i wasn't sure what you meant by this, please clarify.

Line 168: "We determined predictor variables per transect to correspond to chimpanzee nest counts" - maybe "We determined which predictor variables corresponded to chimpanzee nest counts in each transect"?

Line 269-270: "which are considered decayed in measurements of decay rate" - i wasn't sure what you meant by this, please clarify.

Line 271: should be "and steep slopes accounted for 56%.."

Line 282: I'm assuming that, although you found a significant positive correlation between ruggedness and diversity of tree species, the correlation coefficient was not sufficiently high for you to exclude one of the terms, but please clarify this.

One other small thing: I found sections of the paper a bit harder to read because of the frequency of abbreviations in the text. Since this will appear online it seems spelling out things like Greater Mahale Ecosystem, Total Basal Area, etc shouldn't hurt you and will prevent the reader from scrolling back and forth trying to remember what each abbreviation means.

Line 431: You note elsewhere that there was minimal human disturbance in the park, but could chimpanzee preference for rugged terrain also be related to the avoidance of human disturbance?

Line 443: I encourage the authors to avoid causal language when interpreting the results of correlational analyses (i.e. maybe use "associated with" rather than "influenced").

6. PLOS authors have the option to publish the peer review history of their article (what does this mean?). If published, this will include your full peer review and any attached files.

Reviewer #1: No

Reviewer #2: No

---

## [Author Response · Author response to Decision Letter 0]

21 Jan 2021

3.We note that [Figure(s) 1] in your submission contain map images which may be copyrighted. All PLOS content is published under the Creative Commons Attribution License (CC BY 4.0), which means that the manuscript, images, and Supporting Information files will be freely available online, and any third party is permitted to access, download, copy, distribute, and use these materials in any way, even commercially, with proper attribution. For these reasons, we cannot publish previously copyrighted maps or satellite images created using proprietary data, such as Google software (Google Maps, Street View, and Earth). For more information, see our copyright guidelines: http://journals.plos.org/plosone/s/licenses-and-copyright.

The figure in question includes Landsat 8 imagery. USGS/NASA, who have made these data available, state that “there are no restrictions on Landsat data downloaded from the USGS; it can be used or redistributed as desired. We do request that you include a statement of the data source when citing, copying, or reprinting USGS Landsat data or images”. In accordance with this statement, the caption for Figure 1 has now been updated citing our data source. 

1. You may seek permission from the original copyright holder of Figure(s) [1] to publish the content specifically under the CC BY 4.0 license. 

Comments to the Author

1. Is the manuscript technically sound, and do the data support the conclusions?

Reviewer #1: Yes

Reviewer #2: Partly

2. Has the statistical analysis been performed appropriately and rigorously? 

Reviewer #1: I Don't Know

Reviewer #2: Yes

3. Have the authors made all data underlying the findings in their manuscript fully available?

Reviewer #1: Yes

Reviewer #2: No

4. Is the manuscript presented in an intelligible fashion and written in standard English?

Reviewer #1: Yes

Reviewer #2: Yes

5. Review Comments to the Author

Reviewer #1: This is a good data set that is well analyzed. I think that with some work it is definitely suitable for publication in PLoS ONE. However, I have two major criticisms that need to be addressed before I recommend publication.

First, the introduction is lacking in terms of setting up the motivation for the study. The authors make a big deal in the intro of the importance of estimating the overall pop density of chimps in Mahale Park as a benchmark for future conservation efforts. This is reinforced by Table 1, where the estimate densities of other important chimp populations. I was thus led to believe that this was going to be a major goal of the paper. But when it comes to the actual results, all we get is this: Line 303: “Our results

estimate chimpanzee density at 0.23 ind/km2 (0.16 – 0.35 95% CI) across all MMNP, but estimates varied significantly among sites (F2, 12= 58.23, P < 0.001), ranging from 0.09 – 3.43 ind/km2 (Table 306).” This result isn’t even mentioned in the discussion. Do the authors really believe this 0.23 ind/km2 (0.16 – 0.35 95% CI) estimate is robust enough to serve as a benchmark to evaluate the effectiveness or lack thereof of conservation measures at Mahale Park. If yes, explain why and defend that position; if not, drop this angle from the intro.

The second major goal of the study, determining the specific ecological predictors associated with chimp density at Mahale Park, also lacks a good set up. Lines 92-95: “We predicted chimpanzee density to be higher in areas with 1) greater fruit abundance and diversity, 2) high topographic heterogeneity, and 3) more evergreen forested vegetation (includes all available forested vegetation types, i.e., riparian, lowland, and montane forests).” I think the authors need to spend some more time in their introduction setting up the motivation for these predictions. Science is about filling gaps in knowledge, setting up tests that make us favor specific hypotheses/theories over others. The authors don’t do a great job explaining how the answers to these specific questions fit into some bigger debate. Basically, what is at stake here with the specific answers to these questions? What previous research and debates are being built upon, and how will the specific answers given in this paper contribute to the resolution of those debates? There is a little bit of this in the discussion, but those sorts of things should be raised more in the intro. As of now, the intro is too broad, focused much more on the very general issue of trying to convince the reader that it is important to know about ecological predictors of chimp density, and not enough on the more specific issues of what we already know, what we don’t know, and why we need to do X, Y or Z to resolve these issues.

We agree that a single density estimate for the entire park, alone, is robust enough to serve as a benchmark for evaluating the effectiveness of conservations measures within the park. However, our objectives were also to provide density estimates for multiple fixed sites across the park to (1) assess the ecological features that influence density variability and (2) serve as a baseline for future (monitoring) surveys that can help direct efforts inside the park (e.g., repeat surveys that follow the same design as the current study and allow for direct comparisons) and improve our understanding of density patterns outside the park. This allows for more targeted efforts in the future as areas are unlikely to experience homogenous impacts from human disturbance or physical features. We have built on this justification in lines 126-128 and 132-137. 

Regarding your second concern, we have added text that we hope will more clearly demonstrate the knowledge gaps we are trying to fill (lines 64-67, 76-79) and potential contribution to the broader debate on the ecological factors driving chimpanzee density (lines 96-100, 106-112). We hope the changes we have made to our Introduction will clarify these objectives and guide the reader’s expectations more appropriately. 

Finally, my second major criticism is that a lot of the discussion is too far off-topic. In some places it reads more like the authors personal opinions about the best way to conserve chimps in the Mahale area. This is all fine and reasonable stuff, but it doesn’t have a direct relationship to what is analyzed in the paper.

Focus on correlates and not conservation issues unrelated to what I

We have streamlined our discussion section and removed extraneous dialogue to remain within the sphere of our results. 

Line 95: Baseline data on what? On overall population size in Mahale park? Or are you taking about three 3 predictions in the previous sentence?

This refers to acquiring baseline data on chimpanzee density and their habitat at multiple (fixed) sites across Mahale park (line 126-128)

Line 139-141: But you don’t mention anything about human disturbance in the intro. If you are actually going to analyze this in the paper, you need to set it up in the intro. If you are not, you don’t need to describe it in the methods (i.e., you don’t need to describe your entire methodology, but only the bits of it are relevant to the specific questions addressed in the paper).

We have now tried to introduce the concern of human disturbance more upfront in the introduction (lines 49-51, 67-70). While human activity was not included in our model, we believe it is important to mention given the varying levels of human encroachment national parks experience throughout Africa and our assertion that the park can serve as a baseline for the GME region as it experiences little human activity (lines 132-133). We report the level of human activity we observed in our results (lines 297-301).

Line 175: by ‘individual trees’, do you mean ‘different tree species’?

Yes, and we have now re-worded this sentence for clarity (line 210).

2.5 Statistical analyses. You said you had clusters of 5-9 transects per site. Were these transects considered as statistically independent predictors n the model? I worry about autocorrelation, as is suggested by the clustering term. Please clarify.

We have now omitted this confusing term (“clusters”). To address potential spatial autocorrelation, we ran a Moran’s I test, and now discuss these results in the analysis and results section (lines 254-255, 332-333). 

Line 186-187: “We built models to determine the effect of predictor variables on the partially observed true state (i.e., nest density) and detection (i.e., how animals are detected) processes.”

-I don’t understand what these two things you are predicting are exactly, and how they differ, Please explain a bit more.

We have now re-worded this section (lines 246-247). 

Table 3: Do any of the sites correspond to the location of the habituated Mahale chimp groups? I ask because it would be nice to get a sense of the validity of your estimates by knowing how well they compare to the most accurate and precise way of getting densities (i.e., habituating chimps, individually identifying them, and counting them). Many of the density estimates shown in Table 3 have suspiciously narrow confidence intervals.

Yes, site B from the current study and the home range of M group, long-term research community in Mahale, overlap substantially. A discussion of historical and current density estimates for M group and our own has now been incorporated into our discussion (lines 388-392).

Conclusions: Sure, connectivity is great; I am all for connectivity. But what does connectivity have to do with the specific questions you addressed in this paper? You could have also wrote about how poaching, or the chimp pet trade, are bad for chimp populations, and we should reduce those things. But these things too don’t have much of a direct connection with the actual results of your paper.

Line 454-468: Again, I agree with all of these things you write here, but what do they have to do with the results of your paper? In your paper, you found X, when it could have turned out that you found Y. Your discussion should deal with the greater significance/meaning of the fact that you found X instead of finding Y. These things that you are writing about here have little to do with either X or Y.

Line 469-486: Same comment as above.

We have removed much of this text that went beyond the scope of our paper and hope our conclusion now aligns more clearly with the scope of our research and its findings. For example, we highlight findings such as the relative importance of overall food availability in comparison to forest cover in order to stress the value of a landscape approach for conservation practitioners (lines 484-488). 

Reviewer #2: This was a well-written and interesting manuscript that applied sophisticated statistics and extensive field work to the question of estimating chimpanzee densities in different habitats across Mahale Mountains National Park. I believe that the manuscript provides a useful set of results that can inform conservation efforts in the Greater Mahale Ecosystem and within the park itself. I do have several minor issues that I would like to see the authors address.

First, the conclusion section does not refer much to the results from the study. Indeed, there is only one sentence in the conclusion section that refers directly to the results of this study (lines 440-443). Proposed follow-up research is also briefly mentioned (line 469 ff.), but I think it would be valuable explain more clearly in the conclusion how the results of this study, or alternatively the results of the proposed follow-up research, can assist in conservation efforts beyond general statements about biodiversity surveys etc. This would more clearly link the conclusions with the data.

The conclusion section has now been reduced and streamlined, which we hope more clearly explains how these data address knowledge gaps and could be used for future monitoring efforts both within Mahale and outside the park. 

Another issue that I think would strengthen the conclusions in the manuscript: it would be valuable for the reader to discuss the relationship between chimpanzee community range sizes and daily travel distances in Mahale, and the 25 km^2 study sites. You note at the end that you're sampling where chimpanzees sleep, not their entire range (obviously beyond the scope of this project). while 25 square kilometers is a large area, and one would not expect chimpanzees to routinely travel 5 km to nest, I have occasionally seen chimpanzees traveling a kilometer or two at the end of the day to a nesting site, so the question of overall space use vs nest site selection seems quite relevant. Was the size of each study site (5km x 5 km) chosen with the ranging patterns of chimpanzees in mind? It would at least be useful to expand on the difficulties involved in inferring overall space use from sleeping sites. Community range data would be especially useful because it is unlikely that chimpanzees will nest in contested territory between two community ranges, regardless of food species abundance and whether chimpanzees are entering the area during the day to feed (presumably in large groups). I don't know if you have access to those data, and the selection of 13 randomly-chosen sites makes it less likely that an unlucky site placement will bias your results, but it would help the reader and strengthen the conclusions to include some discussion of chimpanzee behavior.

We now explain how we chose our survey site size in our methods (lines 160-161), explaining that we determined a grid cell (survey site) size of 25 km2 according to the home range of M group, the long-term research community in Mahale [12]. Data from sites outside the park show that chimpanzees in less forested areas of the region can display much larger home-ranges, however, we do not believe that any potential differences in home range size for park communities impacts our estimates of chimpanzee density for fixed sites. Yet, we do acknowledge the limitations of our methodology for informing conservation strategies since our methods only utilize nest data. Thus, we advise that additional data on chimpanzee habitat use and range must be taken into account to protect the longevity of the species (lines 488-492).

Methods:

I wasn't totally convinced by the decision to use p values to exclude one of two co-linear terms from the full model. Would it be possible to simply compare alternative models using each of the two terms that were colinear in the full model (steep slopes and ruggedness), rather than excluding one using p values in the full model?

We have now decided not to eliminate highly correlated predictors from being tested in our density models Rather, we retained all predictors for our analysis but coded models accordingly to prevent highly correlated covariates (i.e., steepness and ruggedness) from occurring in the same model (see lines 224-226).

In addition, I had several other small suggestions:

Line 160: "accomplished within 1 - 10 of the chimpanzee data collection" - i wasn't sure what you meant by this, please clarify.

We have omitted this information as it is unnecessary detail.

Line 168: "We determined predictor variables per transect to correspond to chimpanzee nest counts" - maybe "We determined which predictor variables corresponded to chimpanzee nest counts in each transect"?

We have re-worded this line (line 203). 

Line 269-270: "which are considered decayed in measurements of decay rate" - i wasn't sure what you meant by this, please clarify.

We have re-worded this sentence for clarification (line 244).

Line 271: should be "and steep slopes accounted for 56%.."

We have fixed this sentence (line 316).

Line 282: I'm assuming that, although you found a significant positive correlation between ruggedness and diversity of tree species, the correlation coefficient was not sufficiently high for you to exclude one of the terms, but please clarify this.

In our methods section we have now clarified that when coding our models, covariate collinearity is only a concern when covariates are found to be “highly correlated” according to Pearson product‐moment and Spearman rank correlation coefficient test statistics (lines 224-225).

One other small thing: I found sections of the paper a bit harder to read because of the frequency of abbreviations in the text. Since this will appear online it seems spelling out things like Greater Mahale Ecosystem, Total Basal Area, etc shouldn't hurt you and will prevent the reader from scrolling back and forth trying to remember what each abbreviation means.

We have eliminated some abbreviations and opted to write out total basal area and mean basal area within the text. 

Line 431: You note elsewhere that there was minimal human disturbance in the park, but could chimpanzee preference for rugged terrain also be related to the avoidance of human disturbance?

There was insufficient evidence on human disturbance to assess its distribution and whilst interesting, that was not the focus of this paper. Our assumption is that the relative lack of human disturbance in the park suggests that the association between chimpanzee density and rugged terrain is not related to anthropogenic factors. We have added a line in the Discussion recommending that these results are ideal comparisons for future studies outside the park that address anthropogenism – chimpanzee spatiotemporal relationships. 

Line 443: I encourage the authors to avoid causal language when interpreting the results of correlational analyses (i.e., maybe use "associated with" rather than "influenced").

This sentence has now been omitted but similar instances have now been changed to “associated with”.

---

## [Editor Report · Decision Letter 1]

25 Jan 2021

Ecological correlates of chimpanzee (Pan troglodytes schweinfurthii) density in Mahale Mountains National Park, Tanzania

PONE-D-20-30689R1

Dear Dr. Piel,

We’re pleased to inform you that your manuscript has been judged scientifically suitable for publication and will be formally accepted for publication once it meets all outstanding technical requirements.

Kind regards,

Bi-Song Yue, Ph.D

Academic Editor

PLOS ONE

---

## [Editor Report · Acceptance letter]

2 Feb 2021

PONE-D-20-30689R1 

Ecological correlates of chimpanzee (*Pan troglodytes schweinfurthii*) density in Mahale Mountains National Park, Tanzania 

Dear Dr. Piel:

I'm pleased to inform you that your manuscript has been deemed suitable for publication in PLOS ONE. Congratulations! Your manuscript is now with our production department. 

Kind regards, 

on behalf of

Dr. Bi-Song Yue 

Academic Editor

PLOS ONE